# Effects of post oak (*Quercus stellata*) and smooth brome (*Bromus inermis*) competition on water uptake and root partitioning of eastern redcedar (*Juniperus virginiana*)

**Samia Hamati**[1¤]*, **Juliana S. Medeiros**[2], **David Ward**[1]

**1** Department of Biological Sciences, Kent State University, Kent, Ohio, United States of America, **2** Holden Arboretum, Kirtland, Ohio, United States of America

¤ Current address: Department of Biology, University of North Dakota, Grand Forks, North Dakota, United States of America

* shamati@kent.edu

**Data Availability Statement:** The data collected in this experiment is archived in the Open Access

## Abstract

Eastern redcedar *Juniperus virginiana* is encroaching into new habitats, which will affect native ecosystems as this species competes with other plants for available resources, including water. We designed a greenhouse experiment to investigate changes in soil moisture content and rooting depths of two-year-old *J. virginiana* saplings growing with or without competition. We had four competition treatments: 1) none, 2) with a native tree (*Quercus stellata*), 3) with an invasive grass (*Bromus inermis*), and 4) with both *Q. stellata* and *B. inermis*. We measured soil moisture content over two years as well as root length, total biomass, relative water content, midday water potential, and mortality at the end of the experiment. When *J. virginiana* and *B. inermis* grew together, water depletion occurred at both 30–40 cm and 10–20 cm. Combined with root length results, we can infer that *J. virginiana* most likely took up water from the deeper layers whereas *B. inermis* used water from the top layers. We found a similar pattern of water depletion and uptake when *J. virginiana* grew with *Q. stellata*, indicating that *J. virginiana* took up water from the deeper layers and *Q. stellata* used water mostly from the top soil layers. When the three species grew together, we found root overlap between *J. virginiana* and *Q. stellata*. Despite the root overlap, our relative water content and water potential indicate that *J. virginiana* was not water stressed in any of the plant combinations. Regardless, *J. virginiana* saplings had less total biomass in treatments with *B. inermis* and we recorded a significantly higher mortality when *J. virginiana* grew with both competitors. Root overlap and partitioning can affect how *J. virginiana* perform and adapt to new competitors and can allow their co-existence with grasses and other woody species, which can facilitate *J. virginiana* encroachment into grasslands and woodlands. Our data also show that competition with both *Q. stellata* and *B. inermis* could limit establishment, regardless of water availability.

Kent State (OAKS) repository. http://dx.doi.org/10.
21038/hama.2022.1201.

**Funding:** D.W and J.M got funding from NSF. This
work was financially supported by the National
Science Foundation (no. 402109) and the Herrick
Trust, Kent State University.

**Competing interests:** The authors have declared
that no competing interests exist.

## Introduction

Plants have mechanisms to avoid and minimize competition with other plant species [1, 2].
Much competition between plants occurs belowground [3, 4]. An important belowground fac-
tor affecting plant distribution is water availability [3]. For example, plants with different root-
distribution patterns (e.g., root length) might be able to coexist in the same habitat if their
roots can occupy different spaces and are able to exploit resources from distinct areas [5, 6]. A
major mechanism to avoid water stress and enhance the competitive advantage of a plant is its
ability to exploit deeper stores of soil water [7]. Water availability and the competitive abilities
of plants for water uptake has long been recognized as a fundamental driver of vegetation and
landscape structure [8, 9], as it has the potential to shift community composition based on the
water uptake strategies by different species.

The differentiation of root systems has been thought of as a mechanism that reduces
competition and facilitates species coexistence and can be used to understand the spatial
distribution of plants and plant coexistence [10–12]. Competition for water is considered to
occur by water limitation or increase in water demand [13]. However, competition for
water has been less frequently studied than other factors such as nutrients or light [13].
Water availability is a particularly important factor in the survival and establishment of
seedlings and saplings [14, 15]. The impact of soil-moisture depletion on plants is expressed
in diminished plant size, decreased biomass [16], and lower water potential [17]. Moreover,
plants growing in water-limited soils are more prone to hydraulic failure and drought-
induced mortality [17–19]. Water limitation can also affect a plant's nutrient uptake [20].
This will result in a decline in available resources for the different parts of the plant, includ-
ing leaves and roots [21].

Niche differentiation, predicted by the limiting similarity hypothesis [6], is well known to
occur when plant species occupy different environmental niche axes, including soil moisture
and root depth [6, 22–24]. In one of the most prominent theories of niche separation between
trees and grasses, Walter (1939) [10] proposed the *two-layer hypothesis* as an equilibrium
explanation for the coexistence of savanna trees and grasses. Trees and grasses have very dis-
tinct root lengths and different water-use strategies [25]. Walter (1939) [10] proposed that
grasses predominate over trees in savannas because grasses have shallow roots, allowing them
to use water efficiently and take advantage of the available rainfall [25]. In both savannas and
grasslands, grasses are considered superior competitors for water in the upper soil profile [23]
due to their shallow, dense rooting system with high surface area. The presence of trees in a
system can alter soil water content, affecting all associated plants. Trees have access to deeper
water because of their extensive rooting systems [8, 24, 26]. In grasslands, however, trees and
grasses may use overlapping areas in the soil profile, especially while seedlings and saplings are
growing their roots through the soil profile [22, 27]. One aspect that Walter's (1939) two-layer
hypothesis [10] did not focus on is how the roots of two or more tree species interact to allow
these trees to coexist [12]. Woody plants can have different water uptake patterns [28]. A few
studies comparing the depth of water uptake by co-occurring woody species reported some
tree species use only deep soil water, while others can use available water from both shallow
and deep soil layers [11, 24, 29].

Eastern redcedar (*Juniperus virginiana*) is an example of a tree encroaching into grasslands
and savannas, specifically in eastern and central US, including grasslands in the Great Plains
(grassland located in the interior of North America) [30, 31], and midwestern prairies as far
west as Nebraska [32, 33]. *J. virginiana* is the most widespread conifer in the eastern United
States [33]. *J. virginiana* is known to be drought-tolerant [34] and withstands extremes of

drought due to its extensive root system [35], which is both deep and highly branched, potentially giving this tree a competitive advantage in water acquisition over other species [29]. The advantage in terms of drought tolerance of *J. virginiana* makes it a stronger competitor because it needs to take up less water to maintain the same productivity compared to less drought-tolerant species [31, 36–38]. This has important implications for regional ecohydrology [39]. *J. virginiana* appears to be suited for survival in a drought-prone regions [40, 41], provided sufficient soil water while growing deep roots in early life stages which could fundamentally alter the soil ecosystem water-use [39, 40]. However, it is not just woody plants but also grasses [41] may outcompete trees for shallow soil water when their rooting zones overlap [42]. *Bromus inermis* is an invasive deeply rooting rhizomatous perennial grass that is found in many grasslands and old fields [43] and is a potential competitor for *J. virginiana*. An additional potential competitor for *J. virginiana* are *Quercus* spp. *J. virginiana* is encroaching in areas where oak (*Quercus*) trees, such as *Quercus stellata*, are dominant [31, 34]. For example, *J. virginiana* encroached into the Cross Timbers (western edge of the eastern deciduous forest of the U.S.), transforming *Q. stellata*-dominated forests to *J. virginiana* and *Q. stellata* co-dominant woodlands [31]. *Q. stellata* may be a strong competitor for *J. virginiana* because *Q. stellata* also has taproots that can grow throughout the upper several meters of soil in search of moisture and nutrients [44].

Identifying spatial patterns of soil water depletion when plants are growing alone or in competition with other species can help us understand species coexistence and competition for water. Especially during early stages of establishment, *J. virginiana* should transition from shallow roots to deeper roots, which should affect its ability to compete with grasses compared to woody plants [29]. *B. inermis* should rely on water from the top soil layers, thus, *J. virginiana* should experience more water stress when establishing alongside *B. inermis*. However, *J. virginiana* could avoid low water potentials by developing an extensive root system that gives it access to deeper soil moisture. Similarly, water stress for establishing *J. virginiana* should increase with additional competition with *Q. stellata*, but the potential of root partition to ameliorate competition with woody plants is more difficult to predict because they also develop deep root systems. In this study we sought to determine the rooting depth and changes in water availability and water status for *J. virginiana* both growing alone and in competition with *B. inermis* and/or *Q. stellata*. We conducted a greenhouse experiment and used frequency domain reflectometry (FDR) to measure water availability at different depths. We also recorded *J. virginiana* root length, total biomass, and mortality and examined *J. virginiana* water status by measuring midday leaf water potential and relative water content. We further tested Walter's (1939) two-layer hypothesis [10] by focusing on asking whether differentiation in the rooting depth of co-occurring species can allow them to co-exist. We made four predictions based on Walter's (1939) two-layer hypothesis [10]:

1. *J. virginiana* and *B. inermis* interactions will lead to water uptake at different depths.

2. There will be niche separation between *J. virginiana* and *Q. stellata* roots [31].

3. Soil moisture will be consistently low throughout the soil profile when all three species (*J. virginiana*, *Q. stellata*, *B. inermis*) co-occur, leading to lower water potential for *J. virginiana*.

4. We expect that the water requirements for *J. virginiana*, *Q. stellata*, and *B. inermis* will increase with time as the plants grow, leading to more intense water stress in the second year of the experiment.

## Materials and methods

### Ethics statement

The plant materials used in this experiment were purchased from commercial nurseries. *J. virginiana* saplings were obtained from Pinelands Nursery & Supply, New Jersey, *Q. stellata* saplings were obtained from Mossy Oak Nativ Nurseries, Mississippi, and *B. inermis* seeds were obtained from Great Basin Seed company, Utah. The experiment did not involve any endangered or protected species, so no specific permissions were required for the collection and use of plant materials. The experimental research on plants, including the collection of plant material, complied with relevant institutional, national, and international guidelines and legislation. All methods were performed in accordance with the relevant guidelines and regulations.

### Experimental design

We conducted a two-year greenhouse experiment, from September 2017 to August 2019, to analyze changes in soil moisture content, rooting depths, total biomass, water stress, and mortality for *J. virginiana* growing alone and in competition with *Q. stellata*, and/or *B. inermis*. We used 95 L containers (55 cm diameter x 70 cm height) (hereafter *pots*) filled with PRO-MIX® All-Purpose Professional Grower's Potting Mix (BFG Supply). PRO-MIX® has a medium-high moisture retention and a moisture content of 30–50% by weight. We used a completely randomized design in which we grew *J. virginiana* in four treatments (no competition, hereafter *JUVI*, *Q. stellata* competition, hereafter *QUST*, *B. inermis* competition, hereafter *BRIN*, and *Q. stellata* and *B. inermis* competition, hereafter *QUST+BRIN*). We used 16 pots for each treatment. *J. virginiana* saplings (two-year-old, one sapling in each pot, mean initial height ± S.E. = 226±6.6 mm) were planted first, followed by the *Q. stellata* (one-year-old, one sapling in each of the QUST and QUST+BRIN treatments, mean initial height ± S.E. = 319±8.3 mm) and *B. inermis* two weeks after *J. virginiana* was planted, for a total of 64 pots. In the treatments with *B. inermis*, we planted 200 grass seeds/pot with a germination rate of 97%.

We grew the plants in a controlled environment (average temperature = 25°C and four hours of supplemental light in winter). All pots were well watered and irrigated for approximately 10 min every other day through a drip-irrigation system at a rate of 40 ml/min. The same amount of water was supplemented in both years and for all treatment combinations. However, in winter, the plant's growth is slower, and they needed less water to survive. As a result, we irrigated the pots in winter for 5 min every other day through a drip-irrigation system. We monitored soil moisture throughout the experiment to ensure proper drainage to the deeper soil layers. Prior to planting the saplings, we installed a 1 m-long 50 mm-diameter polycarbonate access tube in the middle of every pot to measure moisture content in the soil profile using frequency-domain reflectometry (see below).

### Soil moisture content

We used a Diviner 2000® soil-moisture probe (Sentek Pty Ltd, Stepney, Australia) to obtain moisture content readings every 10 cm in the soil profile [22]. The Sentek system uses frequency-domain reflectometry (FDR). FDR measures soil water content at different depths indirectly by measuring the frequency variations of an electromagnetic pulse propagated into the soil. The difference between the output wave and the return wave frequency is measured to determine moisture content every 10 cm in profile. Factory settings and calibration, to the potting mix used in this experiment and following Sentek recommendations, were used to derive percentage water (content θ%) at each depth. The access tubes, installed before planting, were

used to measure moisture content in the soil profile. Soil moisture content was recorded every other week.

## Plant overall performance

Throughout the experiment, we recorded *J. virginiana* mortality in the four treatments. At the end of the experiment (24 months), all plant material was harvested. *J. virginiana* and *Q. stellata* roots were washed with water to remove soil, extended to their full length, and measured with a ruler to record the maximum root length. We did not measure root length of *B. inermis* due to root breakage during harvesting. To account for the effect of competition on the performance of *J. virginiana*, plants were dried in the oven at 60 ºC for 48 h, then weighed to measure total plant biomass, including root biomass, for *J. virginiana*, *Q. stellata*, and *B. inermis* (S1 Fig).

## Water status of *J. virginiana* saplings

*J. virginiana* leaf relative water content (RWC) was determined for the four treatments at the end of the experiment (August 2019). *J. virginiana* leaf samples were weighed immediately after harvesting (fresh weight), then placed in a beaker with distilled water for 18 h at room temperature and then weighed to determine the saturated weight. The samples were then dried in an oven at 60°C for 24 h to obtain their dry weights. The RWC was calculated using the following formula [45, 46]:

$$RWC = \frac{\text{fresh weight} - \text{dry weight}}{\text{saturated weight} - \text{dry weight}} \times 100 \quad (1)$$

We also measured *J. virginiana* leaf-water potential, at the end of the experiment, between 11:00 am and 1:00 pm using a Scholander pressure chamber, Model 1505D (PMS Instrument Company, New York, USA). Midday water potential represents the time period when the plants are the most active, most water stressed, and have the greatest Vapor Pressure Deficit (VPD) [47].

## Statistical analysis

Data were checked and passed the normality and homogeneity of variance using the Shapiro–Wilk and Levene's test, respectively. A univariate repeated-measures analysis of variance (ANOVA), with α = 0.05, was used to analyze differences in soil moisture content in the four *J. virginiana* treatments among different soil depths and different measurement dates (Sep. 2018, Nov. 2018, Jan. 2019, May 2019, and Aug. 2019).

To determine if the plants used different amounts of water over 2018 and 2019, we also performed an ANCOVA on the soil moisture data for each treatment, with depth as the covariate (continuous variable) and month and *J. virginiana* treatments (JUVI, QUST, BRIN, and QUST+BRIN) as the independent variables. Time periods were defined as the first year for soil moisture data collected in September 2018, and second year for soil moisture data collected in August 2019. To examine possible time shifts in soil moisture content, we used a linear and a cubic regression spline. We also used a LOESS regression to visualize possible trends in soil moisture content between the first (2018) and second year of the experiment (2019) in the four *J. virginiana* treatments.

We ran a univariate analysis of variance (ANOVA) followed by Scheffé *post hoc* tests to analyze differences in total biomass, root length, *J. virginiana* RWC, and midday leaf-water potential in the four *J. virginiana* treatments. Pots with dead *J. virginiana* were excluded from the

analysis. *J. virginiana* sapling mortality in the different treatments was analyzed using a Chi-square ($\chi^2$) test. The LOESS regression and figures were all done in R version 3.6.0 (R Development Core Team, 2019). All other statistical analyses were done in SPSS 26.0 statistical package (IBM SPSS, 2019).

## Results

### Soil moisture content changes in the four *J. virginiana* treatments

Our observations showed that water from the drip irrigation system was percolating down the soil profile and was able to reach the bottom of the pots (70 cm). The soil moisture content varied depending on the treatment (F = 29.07; p < 0.001; Fig 1) and soil depths (F = 270.62; p < 0.001; Fig 1). We did not find an overall shift in soil moisture content among the different measurement periods (Sep. 2018, Nov. 2018, Jan. 2019, May 2019, and Aug. 2019) (F = 1.66; p = 0.14; Fig 1). Soil moisture content among the different measurement periods, however, was dependent on *J. virginiana* treatment combination (Table 1). In addition, we ran both

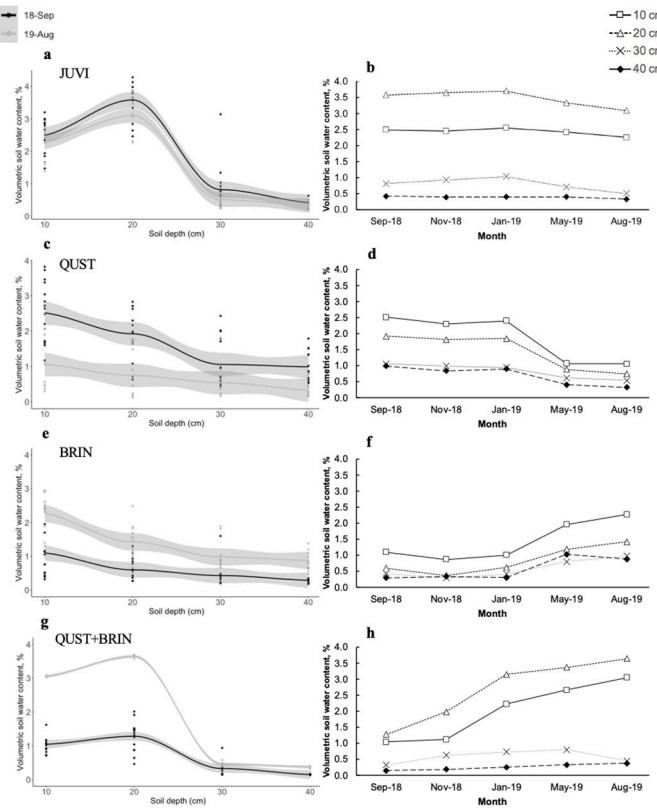

**Fig 1. Relationship between volumetric soil water content (%) and soil depth (cm) in the two years of the experiment (September 2018 and August 2019) and mean soil moisture content at different depths over the experiment (two years).** Volumetric soil water content at different depths in 2018 (18-Sep) and 2019 (19-Aug) for (a) JUVI, (c) QUST, (e) BRIN, and (g) QUST+BRIN. Volumetric soil water content at different depths for the different measurement periods for (b) JUVI, (d) QUST, (f) BRIN, and (h) QUST+BRIN. In panels a, c, e, and g the circles and lines represent moisture content and LOESS regressions for 2018 (black) and 2019 (grey). In panels b, d, f, and h, the square symbols with dashed line represent water moisture data at 10 cm, the triangles and black line represent water moisture data at 20 cm, the circles and dotted line (..) represent water moisture content at 30 cm, and the and diamond long-dashed line (−−) represent water moisture data at 40 cm. JUVI = *J. virginiana* only, QUST = *J. virginiana* growing with *Q. stellata*, BRIN = *J. virginiana* growing with *B. inermis*, and QUST+BRIN = *J. virginiana* growing with both *Q. stellata* and *B. inermis*.

**Table 1. Repeated measures analysis of variance results for the soil moisture content in the four J. virginiana treatments among different measurement dates (Sep. 2018, Nov. 2018, Jan. 2019, May 2019, and Aug. 2019).** JUVI = J. virginiana only, QUST = J. virginiana growing with Q. stellata, BRIN = J. virginiana growing with B. inermis, and QUST+BRIN = J. virginiana growing with both Q. stellata and B. inermis.

| Treatment | Wilks' λ | F | P |
|---|---|---|---|
| JUVI (10 cm) | 0.859 | 0.411 | 0.797 |
| JUVI (20 cm) | 0.502 | 2.479 | 0.111 |
| JUVI (30 cm) | 0.779 | 0.711 | 0.603 |
| JUVI (40 cm) | 0.658 | 1.298 | 0.335 |
| QUST (10 cm) | 0.377 | 4,138 | **0.031*** |
| QUST (20 cm) | 0.331 | 5.055 | **0.017*** |
| QUST (30 cm) | 0.444 | 3.130 | 0.065 |
| QUST (40 cm) | 0.059 | 40.058 | **< 0.001*** |
| BRIN (10 cm) | 0.070 | 30.046 | **< 0.001*** |
| BRIN (20 cm) | 0.094 | 21.686 | **< 0.001*** |
| BRIN (30 cm) | 0.185 | 9.881 | **0.002*** |
| BRIN (40 cm) | 0.052 | 41.244 | **0.005*** |
| QUST+BRIN (10 cm) | 0.003 | 193.201 | **0.001*** |
| QUST+BRIN (20 cm) | 0.008 | 64.947 | **0.015*** |
| QUST+BRIN (30 cm) | 0.009 | 56.741 | **0.017*** |
| QUST+BRIN (40 cm) | 0.002 | 303.736 | **0.003*** |

Reported F values are equivalent F values based on Wilks' λ. Significant values indicated in bold (*).

linear regressions and cubic splines to examine possible time shifts in soil moisture content. All regression lines were significant (p < 0.001). Cubic splines had higher coefficients of determination than linear regressions in all cases (S1 Table).

**Water uptake pattern of *J. virginiana* at different depths.** After comparing our soil moisture data to pots without any plants, *J. virginiana* growing without competitors (JUVI) used more water from the deeper layers of the soil (30 and 40 cm) compared to the shallow depths (10 and 20 cm) (Fig 1a and 1b). We found time progression of soil moisture content between the two years of the experiment (2018 and 2019; Fig 1a). For example, there was significantly more moisture in the soil profile in the first year (2018) than in the second year (2019) (F = 10.29; p = 0.002; Fig 1a; S1 Table), indicating that *J. virginiana* used more water in the second year of the experiment and followed the same water uptake pattern throughout the experiment's timeframe. However, we did not find a significant change in water uptake among the different measurement periods at all depths for JUVI treatment (F = 0.63; p = 0.82; Table 1 and Fig 1b).

**Water uptake pattern of *Q. stellata* at different depths.** When *J. virginiana* grew with the *Q. stellata* (QUST treatment), we found lower soil moisture content at 10 cm and 20 cm depths compared to JUVI treatment (F = 4.47; p = 0.001). Contrastingly, there was no significant change in water uptake at 30 cm and 40 cm compared to the JUVI treatment (F = 0.007; p = 0.8; Fig 1c and 1d). In addition, we found that the soil moisture content during the first year of the experiment (2018) was greater than in 2019 (F = 64.94; p <0.001; Fig 1c), indicating that the *J. virginiana* and *Q. stellata* used more water later in the experiment (S1 Table). In addition, the repeated measures analysis indicated significant changes in moisture content across the different measurement periods. Specifically, there was less soil moisture in May 2019 and Aug 2019 than on other measurement dates at 10, 20, and 40 cm depths (F = 1.96;

p = 0.029). However, there was no significant difference in moisture content at 30 cm during the different measurement periods (F = 1.63; p = 0.176; Table 1, Fig 1d).

**Water uptake pattern of *B. inermis* at different depths.** When *J. virginiana* occurred with *B. inermis* (BRIN treatment), there was a significant decline in soil moisture content at all depths compared to JUVI treatment (Fig 1e and 1f). In contrast to the results of JUVI treatment, the soil moisture content in the BRIN treatment was significantly lower during the first year (2018) than the year of the experiment (2019) (F = 83.89; p < 0.001; Fig 1e). In addition, we found significantly more moisture in the soil profile (the plants took up less water) later in the experiment (May 2019 and Aug 2019) at all depths (F = 2.22; p = 0.011; Table 1, Fig 1f).

**Water uptake pattern of *Q. stellata* and *B. inermis* at different depths.** In the treatment with the most competitors, *Q. stellata* and *B. inermis* (QUST+BRIN treatment), the soil moisture content varied across the experiment. The QUST+BRIN treatment had significantly more moisture in the profile than in the JUVI treatment throughout the experiment at all depths (F = 20.69; p < 0.001; Fig 1g). Moreover, we found that the soil moisture content in 2018 was lower than in 2019 (F = 412.88; p < 0.001; Fig 1g and 1h), indicating that *J. virginiana*, *Q. stellata*, and *B. inermis* used less water later in the experiment (S1 Table). In addition, the repeated-measures analysis indicated a significant change in soil moisture content throughout the experiment at all depths (Table 1).

Examining water uptake at the different depths, our results indicate that *B. inermis* mostly took moisture from the shallow depths (10 cm and 20 cm) due to their shallow roots, whereas *J. virginiana* and *Q. stellata* from the deeper soil layers. There was an overlap in water uptake at the greater depths (30 cm and 40 cm) between *J. virginiana* and *Q. stellata* (Fig 1g and 1h).

## Rooting depths and total biomass

*J. virginiana* root depths and total biomass were dependent on the treatment (F = 35.80; p < 0.001 and F = 12.30; p < 0.001 respectively; Figs 2 and 3). The maximum *J. virginiana* root length was longer than the pot depths (70 cm). However, our observations and data showed no sign that *J. virginiana* saplings were root-restricted or rootbound as they were exploring the

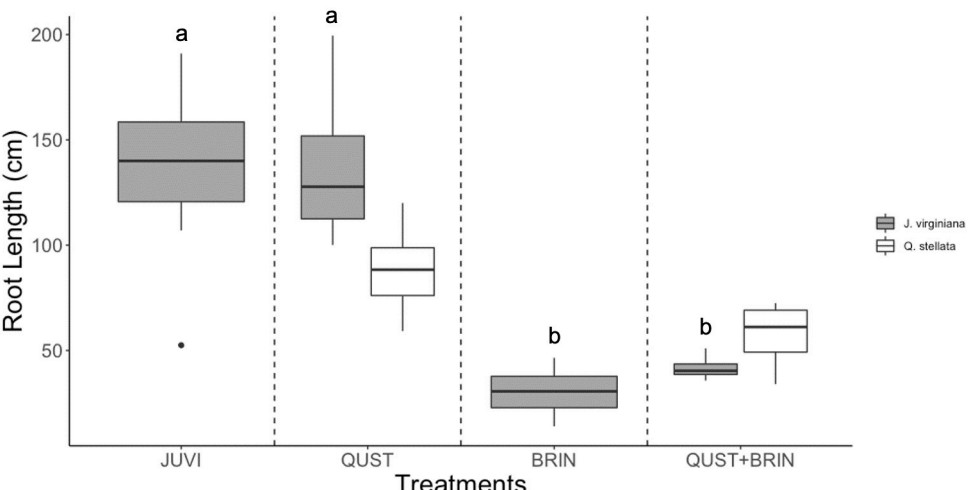

**Fig 2. *J. virginiana* and *Q. stellata* root length (cm) in the four treatments.** *B. inermis* negatively affected *J. virginiana* root length but there was not an effect of *Q. stellata* competition. Note that we did not measure *B. inermis* root length due to root breakage during harvesting. Different letters represent significant differences (Scheffé´ post hoc test). JUVI = *J. virginiana* only, QUST = *J. virginiana* growing with *Q. stellata*, BRIN = *J. virginiana* growing with *B. inermis*, and QUST+BRIN = *J. virginiana* growing with both *Q. stellata* and *B. inermis*.

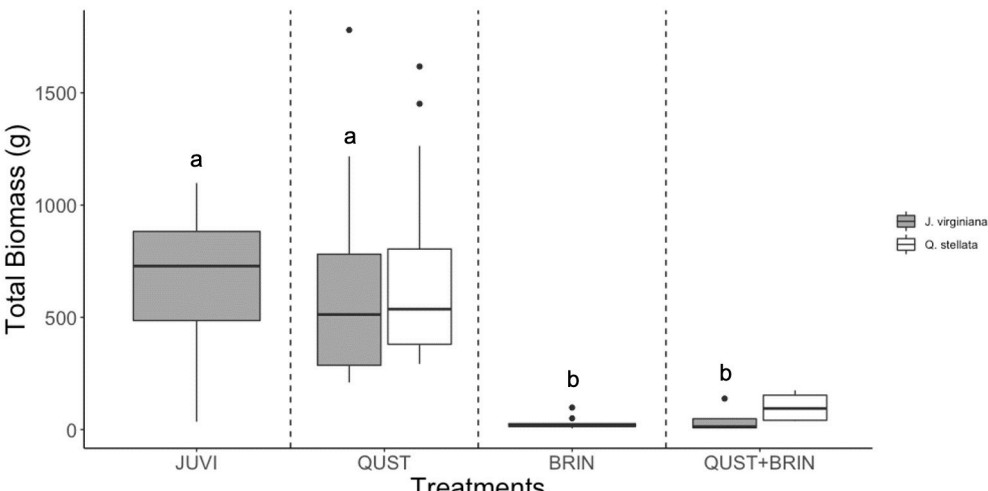

**Fig 3. *J. virginiana* and *Q. stellata* total biomass (g) (above and below-ground) in the four treatments.** *B. inermis* negatively affected *J. virginiana* total biomass. Different letters represent significant differences (Scheffé´ post hoc test). JUVI = *J. virginiana* only, QUST = *J. virginiana* growing with *Q. stellata*, BRIN = *J. virginiana* growing with *B. inermis*, and QUST+BRIN = *J. virginiana* growing with both *Q. stellata* and *B. inermis*.

full volume of the soil. The rooting depth of *J. virginiana* saplings was not significantly affected by *Q. stellata* (F = 0.45; p = 0.83; Fig 2). However, in treatments containing *B. inermis* (BRIN and QUST+BRIN), *J. virginiana* saplings had significantly shorter roots than *J. virginiana* saplings growing in the JUVI or with the QUST treatments (F = 39.34; p < 0.001; Fig 2). Moreover, we found an overlap between *J. virginiana* and *Q. stellata* root lengths in the QUST +BRIN treatment (Fig 2).

The presence of *Q. stellata* did not have a significant effect on *J. virginiana* total biomass (F = 0.01; p = 0.93; Fig 3). However, *J. virginiana* in the JUVI treatment had higher mean total biomass than when grown in the BRIN or QUST+BRIN treatments (F = 25.20; p < 0.001; Fig 3). Overall, *B. inermis* negatively affected *J. virginiana*'s performance.

### *J. virginiana* water status

There was no significant difference in *J. virginiana* RWC (F = 1.64; p = 0.195) between the four *J. virginiana* treatment combinations (Fig 4). However, there was a significant difference in *J. virginiana* water potential between the QUST and BRIN treatments (F = 4.18; p = 0.011; Fig 5).

### *J. virginiana* mortality

*J. virginiana* sapling mortality was affected by the treatment combination ($\chi^2$ = 10.49, df = 3; p = 0.01; Fig 6). We did not find a significant change in *J. virginiana* mortality in JUVI, QUST ($\chi^2$ = 0; p = 1), or with BRIN ($\chi^2$ = 0.8; p = 0.37). There was, however, significantly higher *J. virginiana* mortality when grown in the QUST+BRIN treatment combination ($\chi^2$ = 6.79; p = 0.009; Fig 6).

## Discussion

In order to accurately model and forecast species dynamics and the impact of climate change on ecosystem processes, understanding how plants compete for water is vital. Our experimental design allowed us to investigate changes in *J. virginiana* water uptake and rooting depths to

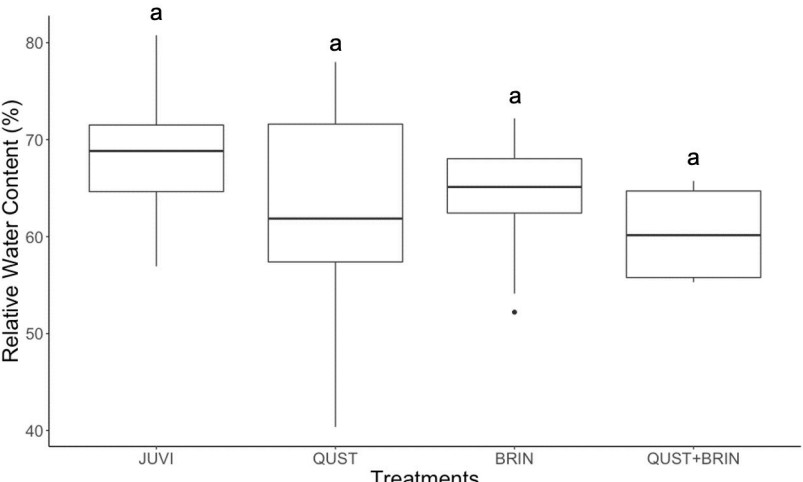

**Fig 4. Relative water content (RWC) for *J. virginiana* grown in four treatments.** There was no significant difference in RWC across treatments, suggesting that it was not water stress that negatively affected the plants in the treatments with *B. inermis*. Different letters represent significant differences (Scheffe´ post hoc test). JUVI = *J. virginiana* only, QUST = *J. virginiana* growing with *Q. stellata*, BRIN = *J. virginiana* growing with *B. inermis*, and QUST+BRIN = *J. virginiana* growing with both *Q. stellata* and *B. inermis*.

competition with an invasive grass (*B. inermis)* and a native tree (*Q. stellata*). We found separation of soil-moisture depletion zones by the different treatment combinations, but our study suggests that factors other than water impact the ability of *J. virginiana* to co-exist with grasses and other woody species. Furthermore, our data suggest that ecohydrological niche separation among coexisting plant species may not occur in the early life stages of woody species and competition for water may change as the plants adopt different plant water-use strategies [48].

### *J. virginiana* water uptake pattern and rooting depth without competitors

*J. virginiana* is known to expand its range and survive in different environments due to its ability to alter its root morphology, allowing for enhanced water uptake, especially during periods

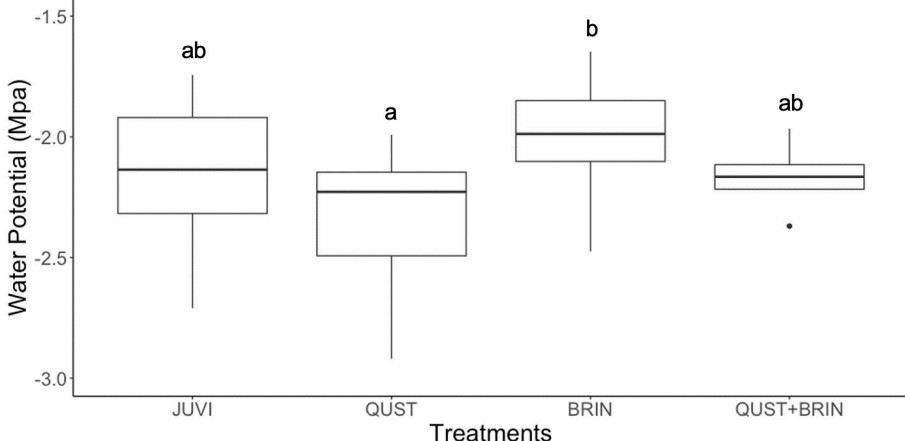

**Fig 5. Midday water potential (MPa) at the end of the experiment for *J. virginiana* grown in four treatments.** There was a significant difference in water potential only between the QUST and BRIN treatments. Different letters represent significant differences (Scheffe´ post hoc test). JUVI = *J. virginiana* only, QUST = *J. virginiana* growing with *Q. stellata*, BRIN = *J. virginiana* growing with *B. inermis*, and QUST+BRIN = *J. virginiana* growing with both *Q. stellata* and *B. inermis*.

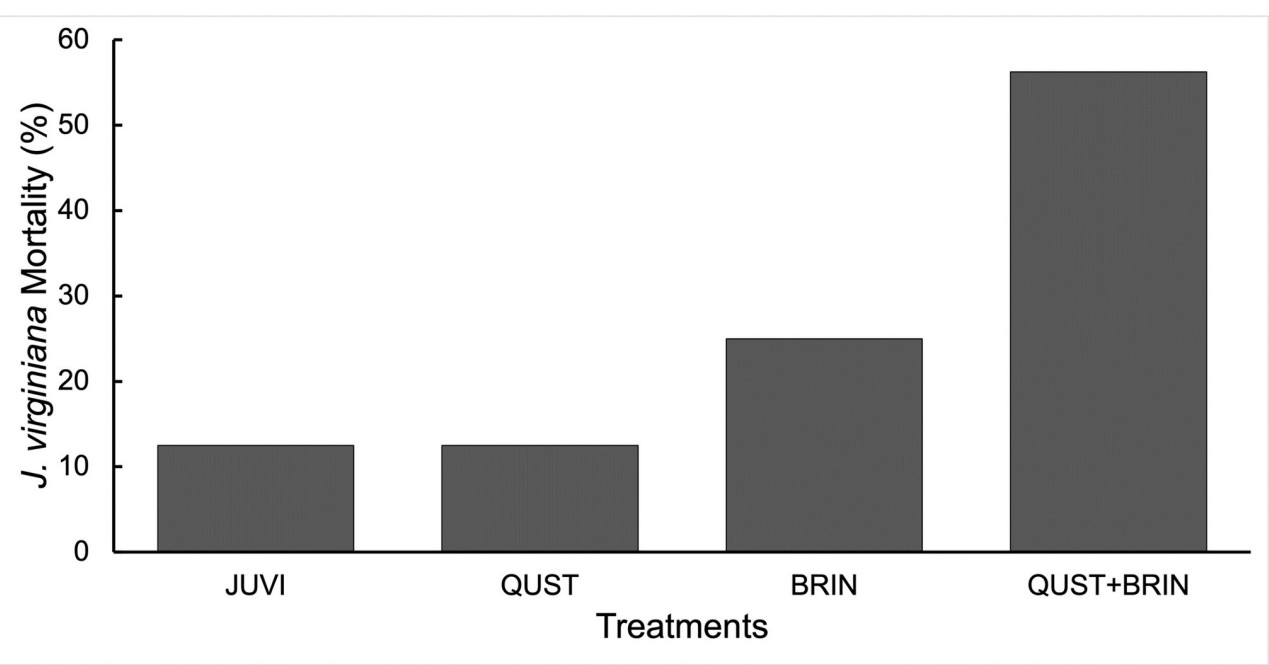

**Fig 6. *J. virginiana* mortality (%) at the end of the experiment in the four treatments.** *B. inermis* negatively affected *J. virginiana* survival. The only significant difference was in the QUST+BRIN treatment. JUVI = *J. virginiana* only, QUST = *J. virginiana* growing with *Q. stellata*, BRIN = *J. virginiana* growing with *B. inermis*, and QUST+BRIN = *J. virginiana* growing with both *Q. stellata* and *B. inermis*.

of low moisture [39]. As predicted, we found that *J. virginiana* growing without competitors used water mostly from the deeper layers of the soil (30 and 40 cm) compared to the shallow depths (10 and 20 cm; Fig 1a and 1b) due to their extensive rooting system (Fig 2). It has been reported that *J. virginiana* roots can reach up to 7 m deep for mature trees [29]. Because we used saplings in this experiment, however, *J. virginiana* roots were not as extensive. Sapling establishment can be strongly suppressed by other plant species, including grasses, than larger trees [49], which makes the sapling stage critical in developing and understanding demographic and encroachment models [49, 50].

### Effect of competition on *J. virginiana* water uptake and rooting depth

One main mechanism in which *J. virginiana* tolerates drought conditions and facilitates its encroachment could be niche partitioning. For instance, differences in rooting depths reduce water competition between trees and grasses [8] and among different tree species [32]. We found that plant interspecific competition had a significant effect on *J. virginiana* water uptake, root lengths, total biomass, and mortality (Figs 1–3 and 6, respectively). Consistent with Walter's two-layer hypothesis (1939, 1979) [10, 51], our results suggest root separation between *J. virginiana* and *B. inermis* roots. *B. inermis* potentially used up more water than *J. virginiana* saplings at shallow depths (10 cm and 20 cm), due to their shallower rooting system. *J. virginiana* likely used the most water at greater depths (30 cm and 40 cm). For instance, it was reported that *B. inermis* roots can reach 1.4 m in depth [52], which is significantly shorter than *J. virginiana* roots. Eggemeyer *et al.* (2009) [29] and Awada *et al.* (2013) [53] have shown that *J. virginiana* competes for soil moisture with grasses in the top layers when water is available. when surface moisture decreases, however, *J. virginiana* shift their water uptake to deeper layers [54]. The FDR method we used to characterize soil moisture depletion does not allow us to

quantify the total amount of water taken up at each depth, rather it gives us a relative measure of soil water depletion across the different depths. Given the interesting patterns described above, water consumption of *J. virginiana* competition should be addressed in future studies to better understand the full impact of water competition on establishment.

We also found root partitioning between the *J. virginiana* and *Q. stellata*, where *J. virginiana* had deeper roots than the *Q. stellata* (Fig 2). As a result, we may infer that *J. virginiana* took up the water from the deeper layers of the soil than *Q. stellata*. Torquato *et al*. (2020) [31] found that *J. virginiana* and *Q. stellata* may be using water from different depths in the oak-dominated Cross Timbers reserve in Oklahoma. In their experiment, Torquato *et al*. (2020) [31] found that *J. virginiana*'s roots occupied the shallower soil depths (45 cm). These results are not consistent with what we found in our experiment because in our study *J. virginiana* had deeper roots than *Q. stellata*. We suggest that the differences between our study and Torquato *et al*. (2020) [31] may be due to ontogenetic differences because they used mature trees whereas we used saplings. The trees may reverse partitioning strategies as a result of ontogeny, but more research is needed to understand the relationship between tree age, size, and water uptake behavior. Plant size, age, and ontogeny may alter the outcome of interspecific interactions and competition for available resources, with seedlings and saplings establishment benefited more than larger individuals [55]. A few studies showed a shift from facilitation during establishment to a competitive effect on adult plant growth and performance [55–59]. This can be related to the fact that larger plants often have greater resource requirements, which increase their competitive impacts. In addition, larger plants are reported to have a lower mortality because they have more extensive root systems [60, 61].

We also examined water uptake at the different depths when *J. virginiana* grew with *Q. stellata* and *B. inermis* (QUST+BRIN). Our results indicate that there was an overlap in water uptake at the greater depths between *J. virginiana* and *Q. stellata*. *B. inermis* has laterally extensive rooting system that mostly took moisture from the shallow depths, potentially leaving less water for *J. virginiana* and *Q. stellata* at the 10 cm and 20 cm depths. As a result, we suggest that the two woody species, *J. virginiana* and *Q. stellata*, invested in belowground development and shared water uptake from the 30 cm and 40 cm layers (Fig 1g and 1h). We also found a decline in *J. virginiana* rooting depths in the QUST+BRIN combination compared to the JUVI treatment (Fig 2) and an overlap between *J. virginiana* and *Q. stellata* roots. This suggests that *Q. stellata* can potentially be considered a major competitor affecting *J. virginiana* performance but perhaps only in the presence of a shallow-rooted species such as *B. inermis*.

## Changes in water uptake over time

When comparing water uptake between the two years for all treatment combinations, we found that *J. virginiana* in the JUVI treatment used significantly more water in the second year (2019) than the first year (2018). As evidenced by the fact that the same amount of water was supplemented in both years but there was more moisture in the soil profile in 2018. This might be attributed to the increase in *J. virginiana* sapling size as they grew older. Similarly, we found that the *J. virginiana* plants competing with *Q. stellata* also used significantly more water in 2019 compared to 2018 (there was a decline in water uptake by the *J. virginiana* and *Q. stellata* in 2019). Unexpectedly, however, in the treatments with *B. inermis* (BRIN and QUST+BRIN), the water uptake by the plants declined in 2019 compared to 2018 (higher soil moisture in 2019 than 2018; S1 Table). This could be related to the negative effect of *B. inermis* on *J. virginiana* and *Q. stellata* survival, growth rate, and total biomass (aboveground and belowground biomass). Consequently, *J. virginiana* and *Q. stellata* had slower growth in 2019 compared to 2018. Another possible cause for the lower water uptakes in the second year of the experiment

could be related to the plants, mainly *B. inermis*, requiring more water to establish their roots and growing at the beginning of the experiment [11], which resulted in lower soil moisture content in the profile. In 2019, *J. virginiana* and *B. inermis* were already established, and their growth rate declined, which ultimately resulted in reduced water requirements and uptake.

## Other factors influencing *J. virginiana* competitive abilities

Grasses can affect woody plant establishment directly, through competition for water, light, or nutrients. Grasses can also have an indirect effect as grass-fueled fires can destroy saplings [62, 63]. Furthermore, sapling emergence, survival, and growth of woody plants are reported to be reduced by grasses [27, 64, 65]. Our results suggest that *B. inermis* had a significant negative impact on *J. virginiana* performance (Fig 3), but the lack of water stress for *J. virginiana* indicates that this was not due to the lack of water availability. Similarly, other studies by Ward (2020, 2021) [41, 66] showed that shade is the most important factor affecting the growth rate and biomass of *J. virginiana*. Interestingly, we found no evidence of the negative effect of *Q. stellata* competition on *J. virginiana* performance even though previous work might suggest that shade by *Q. stellata* would be detrimental for *J. virginiana*. Thus, we suggest our results are likely due to the root partitioning between *J. virginiana* and *Q. stellata*. Furthermore, Torquato *et al.* (2020) [31] found that the soil moisture in pure *J. virginiana* stands was more depleted than in *J. virginiana-Q. stellata* stands, suggesting that *J. virginiana* is a better competitor for water than *Q. stellata*, which may facilitate *J. virginiana* encroachment into oak woodlands. Lastly, we need to recognize that grasses have the opposite effect and control *J. virginiana* encroachment. Thus, the relative importance and effects of grasses and woody species competition needs to be better understood.

## Supporting information

**S1 Fig. *B. inermis* total biomass (g) (above and below-ground).**
(TIF)

**S1 Table. Linear regression and cubic splines equations of the soil moisture content at different depths for the four treatments in the first (2018) and second (2019) year of the experiment.** Cubic splines had higher coefficients of determination than the linear models. All regression lines were significant ($p < 0.001$).
(PDF)

## Acknowledgments

We would like to thank Christian Combs, John Christakis, and Caitlyn Webb for their help in setting up this experiment.

## Author Contributions

**Conceptualization:** David Ward.

**Data curation:** Samia Hamati.

**Formal analysis:** Samia Hamati, David Ward.

**Funding acquisition:** Juliana S. Medeiros, David Ward.

**Investigation:** David Ward.

**Methodology:** David Ward.

**Project administration:** David Ward.

**Supervision:** Samia Hamati, David Ward.

**Writing – original draft:** Samia Hamati, David Ward.

**Writing – review & editing:** Samia Hamati, Juliana S. Medeiros, David Ward.

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
