## [Decision Letter · Decision Letter 0]

31 Aug 2022

PONE-D-22-20878Effects of post oak Quercus stellata and smooth brome Bromus inermis competition on water uptake and root partitioning of eastern redcedar Juniperus virginianaPLOS ONE

Dear Dr. Hamati,

Thank you for submitting your manuscript to PLOS ONE. After careful consideration, we feel that it has merit but does not fully meet PLOS ONE’s publication criteria as it currently stands. Therefore, we invite you to submit a revised version of the manuscript that addresses the points raised during the review process.

We look forward to receiving your revised manuscript.

Kind regards,

Xiao Guo, Ph.D.

Academic Editor

PLOS ONE

Journal Requirements:

"The authors declare no competing interests."

Reviewers' comments:

Reviewer's Responses to Questions

**Comments to the Author**

1. Is the manuscript technically sound, and do the data support the conclusions?

Reviewer #1: Yes

Reviewer #2: No

2. Has the statistical analysis been performed appropriately and rigorously? 

Reviewer #1: Yes

Reviewer #2: Yes

3. Have the authors made all data underlying the findings in their manuscript fully available?

Reviewer #1: Yes

Reviewer #2: Yes

4. Is the manuscript presented in an intelligible fashion and written in standard English?

Reviewer #1: Yes

Reviewer #2: Yes

5. Review Comments to the Author

Reviewer #1: The authors conducted a greenhouse experiment to determine the rooting depth and water uptake across soil layers for ERC both growing alone and in competition with smooth brome and/or post oak. Measurements include root length, water status including midday leaf water potential and relative water content. Water sources and plant competition are very important topics in forest health and mortality under global climate change. However, some issues should be solved in the present version. Because there are few functional traits measured in this study, sufficient context drawn from other publications should be used to strengthen your work. The organization of the manuscript needs to be improved, the Introduction, Result, and Discussion should be shortened and condensed, the Methods should be strongly improved.

The Introduction could be shortened and integrated into four paragraphs, introduction to (1) competition, (2) the two-layer hypothesis, (3) ERC, smooth brome, and post oak, (4) purpose and hypotheses. Moreover, the introduction requires more cohesion and coherence between paragraphs even sentences, and the knowledge gaps should be emphasized.

The Methods should be strongly improved. Some details should be state more clearly. Showing photos related to the experiment can help readers understand your experiment design better.

The Result should be shortened, focus on the brief and scientific description of your results rather than discussing how your results came from. The results could be described one by one according to the figures and tables. Or, you can rearrange the Result according to the logical order of the manuscript.

The Discussion could be integrated into several parts, a subtitle should be given to each part to make the discussion clear. Your predictions should be mentioned and discussed here. In addition, some introductory sentences can be integrated with the Introduction.

Details:

Line 33: Change "very young plants" to "seedlings and saplings".

Line 35–40: Although these are well known, new references need to be added herein. Please see a relevant paper of Liu et al. 2021, Frontiers in Plant Science, doi: 10.3389/fpls.2021.760510.

Line 62–78: What is the root form of ERC, smooth brome, and post oak? In the wild, how long can the root of the three species reach? More root-related functional traits and relevant studies should be introduced here.

Line 112–113: What is the field capacity or saturated moisture content of the PRO-MIX? What are the similarities and differences between PRO-MIX and soil under ERC forest?

Line 121–129: When did the experiment start and finish? Why did the amount of water be controlled? Was it to simulate the local precipitation? In Fig. 1, the volumetric soil water content is lower than 4%, in my point of view, the soil water content in the experiment was so low that it was just like drought treatment. Were the three species faced with drought stress?

Line 131: Too few functional traits were measured in this study. Did you measure the distribution pattern of roots, such as ratio of fine root to thick root, specific fine root area? If not, sufficient context drawn from other publications should be used to strengthen your work.

Line 132: Change "trunk diameter (at root collar)" to "basic diameter". How do you show your basic diameter? It is better to draw a figure to show the height and basic diameter in different periods.

Line 161: Before ANOVA, was normality and homogeneity of variance tested? What statistical methods did you use to test normality and homogeneity? Please state clearly at the beginning of the Statistical analysis. Besides, the critical value (α) of the test should be mentioned.

Line 326–334: This part looks more like an introduction, maybe it is better to be integrated with the Introduction.

Line 388: Here the authors made a comparison between the two years. Soil moisture is not only related to plants, but also to the weather. Was the weather of 2018 growing season similar to that of 2019?

Line 397–399: In my opinion, the larger the plant is, the larger the plant water consumption will be. The authors should deeply discuss the reason why the water uptake by the plants declined in 2019, because that was unexpected.

Line 402: Change "regulate" to "affect".

Line 412: Change "strands" to "stands".

Line 413: Did you pay attention to the functional traits of smooth brome, such as root length, biomass?

Some relevant references below may be helpful to improve the Discussion:

Barbeta A, and Peñuelas J. 2017. Increasing carbon discrimination rates and depth of water uptake favor the growth of Mediterranean evergreen trees in the ecotone with temperate deciduous forests. Global Change Biology 23:5054–5068.

Bréda N, Huc R, Granier A, and Dreyer E. 2006. Temperate forest trees and stands under severe drought: a review of ecophysiological responses, adaptation processes and long-term consequences. Annals of Forest Science 63:625–644.

Epron D and Dreyer E. 1993. Long-term effects of drought on photosynthesis of adult oak trees [Quercus petraea (Matt.) Liebl. and Quercus robur L.] in a natural stand. New Phytologist 125:381–389.

Liu X, Wang N, Cui R, Song H, Wang F, Sun X, Du N, Wang H, and Wang R. 2021. Quantifying Key Points of Hydraulic Vulnerability Curves From Drought-Rewatering Experiment Using Differential Method. Frontiers in Plant Science 12:627403.

Liu X, Zhang Q, Song M, Wang N, Fan P, Wu P, Cui K, Zheng P, Du N, Wang H, and Wang R. 2021. Physiological Responses of Robinia pseudoacacia and Quercus acutissima Seedlings to Repeated Drought-Rewatering Under Different Planting Methods. Frontiers in Plant Science 12:760510.

Liu Z, Liu Q, Wei Z, Yu X, Jia G, Jiang J. 2021. Partitioning tree water usage into storage and transpiration in a mixed forest. Forest Ecosystems 8:72.

Liu Z, Yu X, Jia G. 2019. Water uptake by coniferous and broad-leaved forest in a rocky mountainous area of northern China. Agricultural and Forest Meteorology 265:381–389.

Montagnoli A, Dumroese RK, Terzaghi M, Onelli E, Scippa GS, and Chiatante D. 2019. Seasonality of fine root dynamics and activity of root and shoot vascular cambium in a Quercus ilex L. forest (Italy). Forest Ecology and Management 431:26–34.

Poca M, Coomans O, Urcelay C, Zeballos SR, Bodé S, Boeckx P. 2019. Isotope fractionation during root water uptake by Acacia caven is enhanced by arbuscular mycorrhizas. Plant and Soil 441:485–497.

Zadworny M, Mucha J, Jagodziński AM, Kościelniak P, Łakomy P, Modrzejewski M, Ufnalski K, Żytkowiak R, Comas LH, and Rodríguez-Calcerrada J. 2021. Seedling regeneration techniques affect root systems and the response of Quercus robur seedlings to water shortages. Forest Ecology and Management 479:118552.

Reviewer #2: I think this is a well written paper with some interesting results about rooting depth and competitive water extraction. The biggest criticism I have of this paper is the methods used to identify which species was extracting water from specific locations in the soil profile. I think what this design can say is where the water was extracted in the soil profile and not by whom. It is not easy to predict the performance of plants growing in mixtures compared to when they are going alone and root morphological and physiological responses to neighbors may differ from growth in the control, so more direct information is required. Additionally, methods like tracers in soil layers are some useful methods to assess by whom the water is extracted at particular soil layers. I would suggest revising the argument without focusing on the ability to say which species accessed water in which soil layers. Below are additional comments on the manuscripts.

Line 6 ERC is colloquial. Rather call it J. virginiana and target species are usually spelled out

Line 13-14: Is it root overlap & soil water extraction at the same depth? Resources use may not overlap even if roots are?

Line 37 requires a citation.

Line 38: Change “Reduce” to “reduces”

Line 40 citation needed.

Line 53: Is this case about tree seedlings?

Line 66: change "are" to "is"

Line 105: "second" to "the second"

Line 115: Towards reproducibility of this work: where are the species from? Were the seeds or seedlings purchased or wild collected? When? How were the seedlings propagated for this experiment? About how old were they upon planting?

Line 116 how much time in between planting additional species?

Line 125: Change "plants" to "plant's"

Line 144 say a little more about FDR

Line 146: Where were the access tubes in relation to the plants? Thus, how do you know that the individual species had differential water extraction depths?

Line 182: "Pots with dead ERC were excluded from the analysis." Already said in the methods.

Line 189 Report statistics here

Line 192 the figure 1 legend text is quite long

Line 221 The "ERC" subtitle could be more informative

Line 232 more informative subtitle

Line 234-237: I think what this design can say is where the water was extracted in the soil profile and not by whom. It is not easy to predict the performance of plants growing in mixtures compared to when they are going alone and root morphological and physiological responses to neighbors may differ from growth in the control. Additionally, methods like tracers in soil layers are some useful methods to assess by whom the water is extracted at particular soil layers.

Line 240 "changes" to "changes in"

Line 246 more informative subheading

Line 252-253: Save interpretation for the discussion "This means that the ERC and smooth brome required less water later in the experiment (S1 Table)."

Line 284: The sheer density differences between the grass and the non-target tree has to play a key role here towards growth suppression.

Line 353: This is similar to expectations of the limiting similarity hypothesis that was coined after this work. See Fargione & Tilman 2005: Niche differences in phenology and rooting depth promote coexistence

with a dominant C4 bunchgrass. Oecologia

Fig 2. legend - make a note that not all species root length was measured in the treatments

Fig. 2 say "different letters" rather than "different symbols"

6. PLOS authors have the option to publish the peer review history of their article (what does this mean?). If published, this will include your full peer review and any attached files.

Reviewer #1: No

Reviewer #2: No

---

## [Author Response · Author response to Decision Letter 0]

24 Oct 2022

We thank the Editor and Reviewers for their comments and suggestions to improve our manuscript.

We rearranged, modified, and shortened the different parts of the manuscript, including the Abstract, Introduction, and Discussion. We only included information pertinent to the interpretation of the results of our experiment.

We agree with the Reviewers that our design does not allow us to say from which depth water was extracted in the soil profile and by whom. Therefore, we modified our language in the Results and Discussion to predict and infer shift in water uptake at different depths, between treatments, and over two years.

We replaced ERC, post oak, and smooth brome with J. virginiana, Q. stellata, and B. inermis, respectively. In addition we replaced the ERC treatment with JUVI, ERC + post oak with QUST, ERC + smooth brome with BRIN, and ERC + post oak + smooth brome with QUST+BRIN.

Finally, we ensured that our manuscript meets PLOS ONE's style requirement.

---

## [Decision Letter · Decision Letter 1]

16 Nov 2022

PONE-D-22-20878R1Effects of post oak Quercus stellata and smooth brome Bromus inermis competition on water uptake and root partitioning of eastern redcedar Juniperus virginianaPLOS ONE

Dear Dr. Hamati,

Thank you for submitting your manuscript to PLOS ONE. After careful consideration, we feel that it has merit but does not fully meet PLOS ONE’s publication criteria as it currently stands. Therefore, we invite you to submit a revised version of the manuscript that addresses the points raised during the review process.

We look forward to receiving your revised manuscript.

Kind regards,

Xiao Guo, Ph.D.

Academic Editor

PLOS ONE

Journal Requirements:

Reviewers' comments:

Reviewer's Responses to Questions

**Comments to the Author**

1. If the authors have adequately addressed your comments raised in a previous round of review and you feel that this manuscript is now acceptable for publication, you may indicate that here to bypass the “Comments to the Author” section, enter your conflict of interest statement in the “Confidential to Editor” section, and submit your "Accept" recommendation.

Reviewer #1: All comments have been addressed

Reviewer #2: (No Response)

2. Is the manuscript technically sound, and do the data support the conclusions?

Reviewer #1: Yes

Reviewer #2: Yes

3. Has the statistical analysis been performed appropriately and rigorously? 

Reviewer #1: Yes

Reviewer #2: Yes

4. Have the authors made all data underlying the findings in their manuscript fully available?

Reviewer #1: Yes

Reviewer #2: Yes

5. Is the manuscript presented in an intelligible fashion and written in standard English?

Reviewer #1: Yes

Reviewer #2: Yes

6. Review Comments to the Author

Reviewer #1: I have read the new version of the manuscript. Authors have rearranged, modified, and shortened the manuscript and responded positively to my comments. Thank the authors for receiving the comments constructively and comprehensively revising the manuscript. Besides, I only have some minor points that need to be clarified and/or corrected.

Line 4: It is better to enclose scientific names in brackets, "Effects of post oak (Quercus stellata) and smooth brome (Bromus inermis) competition on water uptake and root partitioning of eastern redcedar (Juniperus virginiana)".

Line 30: Change "two-year old" to "two-year-old".

Line 166: Change "two-years old" to "two-year-old".

Line 167: Change "one-year old" to "one-year-old".

Line 172: Delete the latter "were".

Line 191: Change "were" to "was".

Line 224: Change "different a different amount" to "different amounts".

Line 234: Change "root-length" to "root length".

Line 388: Change "suggest" to "suggests".

Line 466: Change "indicate" to "indicates".

Reviewer #2: The changes from the previous review have helped the manuscript, but there's still more to clarify. Please see my suggestion below.

Line 29-31: treatment summary isn't clear

Line 97: change "withstand" to "withstands"

Line 113: change "have" to "has"

Line 118: change "their" to "its"

Lines 128 - 143: These hypotheses need to be simplified, made more direct and falsifiable.

For example, hypothesis one should read something like: "J. virginiana and B. inermis interactions will lead to water uptake a different depths." The part that reads "because B. inermis should rely on water from the topsoil layers and J. virginiana will invest in belowground development and will extend their roots to get to the deeper water in the soil profile." is something that should be expected given a well-fleshed out introduction, or it should be covered in the discussion if these are the resulting outcomes.

Same for all of the hypotheses.

line 185: change "measure" to "measures"

Line 224: remove "different"

Line 238: towards reproducibility: clarify which tests were run in R and which in SPSS

Line 251-252: Give this sentence a little more care and indicate the p value for each.

Table 1: tables and figures should be interpretable without reference to the text. So, the treatment acronyms require explanation. Clarify that QUST+BRIN contains 3 species

Line 277: save "As predicted" for the discussion

line 278-280: re: "likely" - they either did or they didn't; and "due to their extensive rooting system" this interpretation is better left for the discussion since this wasn't tested directly.

Line 292 -294: Save for the discussion "We can likely attribute this decline in moisture content at the shallow depths to water uptake by Q. stellata because there was no significant change in water uptake at 30 cm and 40 cm compared to the JUVI treatment"

Line 307-308: save interpretation for the discussion "We can attribute this decline to the increase in water demand, mainly to the presence of B. inermis."

Line 309: "in contrast" and "dissension" are redundant

Line 329: save for the discussion "potentially due to the overlap in J. virginiana and Q. stellata rooting depths."

Line 382: One of the previous reviewers suggested adding subheadings to the Discussion and I think this is needed for clarity.

Line 384: rephrase "co-exist for water", e.g., "compete for water"

Line 407-411: this is unknown

Line 408: It sounds like you have the data to graph how much water was extracted from the different soil depths to better make this conclusion - like a density plot or a violin plot.

line 463: clarify "indirectly as fuel for fires" to you mean the grass-fueled fires destroy saplings?

7. PLOS authors have the option to publish the peer review history of their article (what does this mean?). If published, this will include your full peer review and any attached files.

Reviewer #1: **Yes: **Xiao Liu

Reviewer #2: No

---

## [Author Response · Author response to Decision Letter 1]

14 Dec 2022

We thank the Editor and the Reviewers for their comments and suggestions to improve our manuscript.

We have made the recommended and suggested changes to our manuscript including simplifying the hypotheses and adding subheadings to the Discussion.

---

## [Editor Report · Decision Letter 2]

20 Dec 2022

Effects of post oak (Quercus stellata) and smooth brome (Bromus inermis) competition on water uptake and root partitioning of eastern redcedar (Juniperus virginiana)

PONE-D-22-20878R2

Dear Dr. Hamati,

We’re pleased to inform you that your manuscript has been judged scientifically suitable for publication and will be formally accepted for publication once it meets all outstanding technical requirements.

Kind regards,

Xiao Guo, Ph.D.

Academic Editor

PLOS ONE

Additional Editor Comments (optional):

All the comments have been fully addressed. 
---

## [Editor Report · Acceptance letter]

5 Jan 2023

PONE-D-22-20878R2 

Effects of post oak (*Quercus stellata*) and smooth brome (*Bromus inermis*) competition on water uptake and root partitioning of eastern redcedar
(*Juniperus virginiana*) 

Dear Dr. Hamati:

I'm pleased to inform you that your manuscript has been deemed suitable for publication in PLOS ONE. Congratulations! Your manuscript is now with our production department. 

Kind regards, 

on behalf of

Dr. Xiao Guo 

Academic Editor

PLOS ONE